# The Determinants of Tourist Preferences of the Mediterranean Region in Turkey

**Emrah Sitki Yilmaz** [1,*,†] **and Mehmet Aytekin** [2,†]

1    Department of Marketing, Gaziantep University, 27000 Gaziantep, Turkey
2    Production Management, Gaziantep University, 27000 Gaziantep, Turkey; aytekin@gantep.edu.tr
*    Correspondence: esyilmaz@gantep.edu.tr; Tel.: +90-342-317-2036
†    These authors contributed equally to this work.

**Abstract:** Moving marketing strategies, which are changed through digital channels, into the social media environment, has led to changes in customer preferences in the tourism sector and is affected by the comments made in social media. Despite the fact that numerous studies have examined the impact of online customer comments on purchasing behaviour, most of these studies have used descriptive statistics and have ignored the empirical estimations. This study is aimed at examining the influence of the criteria obtained by content analysis on the star ratings given to the hotels based on the comments about the hotel on the TripAdvisor site. In this context, Mediterranean region hotels, which hold an important place in the tourism sector of Turkey, have been viewed, and the 25 hotels with the highest number of comments on the TripAdvisor site were selected, and 9000 comments from these hotels as well as the topic of the study were analysed with panel data methodology. As a result of the analyses made, it was determined that the criteria of the location and the accessibility of the hotels, the food and beverage concept presented by the hotels, the quality of the personnel–service, the cleaning elements, and the entertainment–animation programs were influential on the star ratings given to the hotels. It was also found that the architectural structure of the hotels and the other customers' recommendation criteria are less important than the star ratings given to the hotel when compared to other specified criteria.

**Keywords:** social media; TripAdvisor; Mediterranean Region; consumer comments; content analysis

## 1. Introduction

The increasingly competitive environment and the strengthening of social networking sites have led to the need to keep the marketing strategies of companies at their best in different circles. The impact of social networking sites and digital channels, according to traditional word of mouth (WOM), has a worldwide influence and pre-emption (Deng et al. 2014). In its simplest terms, the WOM concept, defined as the presence of any brand, product, or service among the daily chats, has become a great opportunity and source of information for interpretations, opinions, and suggestions, as digital applications and communication platforms grow rapidly, and consumers increase their bidirectional exchange of information about brands, products, or services. With the developing Web 2.0 applications and the digital channels that have emerged, the WOM concept has turned into the e-WOM concept, and this concept has revealed the impact of social networking sites (Ladhari and Michaud 2015). E-WOM has a very important place in election behaviour in order to reduce uncertainty and perceived risks when consumers make purchasing decisions. Consumers can easily access other internet users through various websites and social networking sites, regarding products and services they consider to be or are not interested in, and they can easily reach the comments, opinions, and recommendations they need (Litvin et al. 2008; Godes and Mayzlin 2004; Lee and Youn 2015). This has further increased the

importance of e-WOM on information acquisition and purchasing decisions, particularly in the tourism and accommodation sectors (Liu and Park 2015). This rapid progress in information technology has caused a global change in the tourism and accommodation sector (Buhalis and Law 2008). The fact that consumers reach all of the information about the hotel quickly, and the fact that businesses centre their consumers on these organizations, are the most important points of this change globally (Neuhofer et al. 2012). This change offers great opportunities not only for the consumers, but also for the hotels, which are one of the important points of the tourism sector. The fact that hotels in the sector exchange active information with customers, allows for access to opportunities such as the evaluation of customers' comments, opinions, and suggestions; the creation of trust and awareness; restructuring of customer experience; the development of brand perception; broad customer knowledge; and reaching feedback to develop new product or new service are among the most important benefits provided by social networking sites and digital platforms (Baird and Parasnis 2011; Sigala 2012; de Rosario et al. 2013).

Along with these advantages, digital travel platforms such as TripAdvisor regarding the purchase of products, services, and accommodation in the tourism sector, and the comments, opinions, and recommendations shared by these platforms make it more important (Park and Kim 2008). Consumers are more confident in consumer comments in preference processes that require high interest. The positioning of the TripAdvisor platform at such a crucial point in decision making abut hotels is due to the customer's emphasis on the customer comments on these and similar platforms in comparison to criteria such as the price, location, and performance in hotel preferences (Gretzel and Yoo 2008).

With the advancement of information technology, the world economy has become one of the fastest growing and expanding sectors of the tourism industry, and is located in a very important point for Turkey. Becoming a centre of attraction for tourists thanks to its geographical location among the Mediterranean countries, Turkey is considered as one of the tourism capitals, especially because of its location, natural beauties, the facilities offered by the hotel operators, and the relatively low-cost tourist facilities.

In this respect, the aim of this study is to understand how much the increased competition among the hotels in the Mediterranean Region, which is one of Turkey's most important tourism destinations, together with developments in the field of technology and competition, reflection on consumer comments, and the criterion determined by the content analysis made by way of these interpretations affect the potential customers' hotel preferences. According to the comments made by users on the TripAdvisor site, the first 25 hotels belonging to the Mediterranean region have been identified, and a total of 9000 comments on these hotels have been analysed for their content and criteria, in order for the inclusion of specific subtitles to be determined. The effects of these criteria on customers' hotel preferences were then examined for a three-year period via panel data analysis. It is expected that the determination of these effects will make important contributions both to the hotels in the sector and to the literature.

## 2. Literature Review

One of the world's most popular travel sites, TripAdvisor, was founded in early 2000s in the United States of America (Law 2006). This site aims to publish an e-consulting service without online booking, aiming to holistically publish the contents created by users for accommodation and travel destinations (Vásquez 2011). With thousands of hotels, interesting and historic sites, cities, restaurants, and traveller photos, this site is also a channel where millions of consumers look for travel plans (Law 2006).

More than 80% of today's travellers visit sites like TripAdvisor, and such sites affect or change consumers' travel decisions (Briggs et al. 2007). In addition, 77.9% of potential customers searching for information on online travel sites are examining other consumer shares, and indicate that these shares are helpful in determining hotel preferences (Gretzel and Yoo 2008). This led to the conclusion that the

potential of the internet-based positive or negative interpretations made for the tourism sector in the literature is very strong and wide, and that they have changed consumers' minds (Briggs et al. 2007; Chung and Buhalis 2008; Miguéns et al. 2008; O'Connor 2008; Yoo and Gretzel 2009). In this context, a large number of studies have been carried out by many authors regarding the tourism sector in relation to social media and digital channels (Law 2006; Jeong and Jeon 2008; O'Connor 2008; Barcala et al. 2009; Law et al. 2009; Simms 2012; Lu and Stepchenkova 2012).

The tourism sector, one of the sectors that frequently needs to be used in terms of the information intensity of social media, is especially influential in influencing potential customers through the use of sites such as TripAdvisor (Jeong and Jeon 2008; Zheng et al. 2009; Sparks and Browning 2010). Digital channels, such as TripAdvisor, provide potential clients with useful information, as well as giving a better administrative understanding of the dynamics in the industry (Limberger et al. 2014).

A number of studies have been carried out in which the criteria affecting consumer preferences are determined based on the comments and information from the TripAdvisor website, one of the leading social media digital channels for the tourism sector. In studies that were done by Jeong and Jeon (2008) and Barcala et al. (2009), as well as ones that reveal the relationship between the service provided and the customer expectations, the results of the services provided by the guests and the overall satisfaction of the customers are based on criteria such as service quality and room prices (Jeong and Jeon 2008). In the study conducted by Barcala et al. (2009), it was seen that customers focused more on hotel factors such as price, location, number of stars, and service.

The reasons for the utilization of TripAdvisor-specific studies are stated as the evaluation of the general satisfaction level, as well as the specification of the evaluation criteria separately and the determination of which of these criteria has the strongest relationship with overall satisfaction (Limberger et al. 2014). The impact of comments made on TripAdvisor on previous purchases made on the site has been examined, but the factors affecting consumer decisions have not been discussed. In the 2010 and beyond, the following studies about the factors influencing the decisions of the customers were examined, such as the comments made about the hotel via the TripAdvisor site (Barriocanal et al. 2010); price-performance, rooms, food–beverage, and physical structure (O'Connor 2010); rooms, personnel–service, food–beverage, and price-performance (Stringam et al. 2010); hotel architecture, cleanliness, staff-service, rooms, food–beverage, and location (Limberger et al. 2014); price, location, rooms, cleanliness, and staff-service; and location, cleanliness, and overall hotel score criteria, have come to the forefront.

In this study, the influence of the criterion obtained from the comments made about the hotels on the TripAdvisor site on the star rating given to the hotels is examined. Therefore, the following hypothesis is tested in this study.

**Hypothesis H1 (H1).** *The criteria obtained from comments on hotels on the TripAdvisor site have a positive effect on the star ratings given to the hotel.*

## 3. Empirical Model, Data Set, and Methodology

In this section, the preference reasons of the hotels in the Mediterranean Region, located on the southern coastline of Turkey, have been examined, and the empirical model used in empirical analysis, the determined data sets, the way in which these data sets are obtained, and the used econometric methods have been introduced.

In light of this information, firstly considering the literature and theoretical information about the subject, the empirical model, which was established for the purpose of examining the effects of seven independent variables that are considered to be important in hotel preferences by the users of thetripadvisor.com.tr site on the star ratings given to the hotel have been introduced. Consequently, information about the way of obtaining the data sets belonging to the mentioned variables is given. Finally, the panel data methods to be used in the empirical analyses are introduced.

### 3.1. Empirical Model

The empirical model panel version, which was created to examine the effects of the criteria on the hotels' star rating, which the customers consider in the hotels' preference in the study, is as follows:

$$lnSP_{it} = \beta_0 + \beta_1 lnARC_{it} + \beta_2 lnGC_{it} + \beta_3 lnFB_{it} + \beta_4 lnPS_{it} + \beta_5 lnEA_{it} + \beta_6 lnLOC_{it} + \beta_7 lnREC_{it} + \varepsilon_{it} \quad (1)$$

Among the variables in the equation, *lnSP* is the natural logarithm of the hotel's star point average, *lnARC* is the natural logarithm of the structure given to the architecture of the hotel, *lnGC* refers to the natural logarithm of the structure given to the general cleaning services of the hotel, *lnFB* means the natural logarithm of the food and beverage quality given to the hotel in concept, *lnPS* is the natural logarithm of the hotel's personnel and service quality, *lnEA* is the natural logarithm for the hotel's entertainment and animation services, *lnLOC* refers to the natural logarithm of the importance given to the location of the hotel, and *lnREC* refers to the natural logarithm of the level of recommendation of the hotel by other users. In addition, *i*, *t*, and $\varepsilon_{it}$ in the equation represent the cross section (the hotel), time period, and error term, respectively. The variables in the model are logarithmically included in the model to prevent possible problems based on dimensional distortions.

### 3.2. Data Set

In the study, the hotels located in the Mediterranean region were studied separately. The 25 hotels, among 4380 hotels in the region, with the highest number of comments on TripAdvisor's site, were included in the sample. When the data sets were prepared, the monthly average of the comments made for each hotel and the star points given were included in the model, and the "2015m1–2017m12" observation range was examined. The numbers of the hotels identified in the data set and the comments about the hotels are given in Appendix A.

A total of 9000 comments from the total of 25 hotels were that chosen for the selected region were analysed, and as a result of the content analysis made, seven different criteria with various subtitles were determined from the comments they made.

Hotel architecture criterion, which is the first of the criteria; includes the hotel architecture and the general situation of the hotel, the size of the rooms, the usefulness of the rooms, the usefulness and novelty of the properties and fixtures of the hotel and the rooms, swimming areas (sea, pool, aqua park, beach, etc.), social areas (restaurants, bars, spa, sauna, hammam, amphitheatre, gym, mini club, etc.), and easy access to all points within the hotel. The second criterion, the cleanliness criterion, includes sub-headings such as the general cleanliness of the hotel, the cleanliness of the rooms (bath, toilet, bed sheet, etc.), the cleanliness of the environment and the cleanliness of swimming areas (sea, pool, aqua park-beach, etc.), the cleanliness of social areas (restaurants, bars, spa, sauna, hammam, amphitheatre, mini club etc.), and the cleanliness of all of the equipment used throughout the hotel. The third criterion, the food–beverage criterion, includes food and beverage variety, food and beverage flavour, freshness, brand quality of food and beverages, food and beverage service, easy access to food and beverage areas, children's restaurant and children's menu, the satisfaction level related to bars all around the hotel, and minibars in rooms in terms of variety and quality. The fourth criterion, the staff-service one, consists of sub-topics such as interest and professionalism of the staff, service speed and quality, problem solving, friendliness and kindness, check-in/check-out, room service and lobby services, welcoming, farewell, and bell boy service. The fifth of these criteria, entertainment–animation criterion, includes entertainment (party, concert, DJ performance, live music, etc.), animation (sea, pool, aqua park, beach, etc.), mini clubs (kids' activity and child care), spa, sauna, hammam, amphitheatre, activities (parachute, balloon, jet ski, games, competitions, etc.), and local tours and excursions. The sixth criterion, the location–transportation criterion, includes the proximity of the hotel to the locations (airport, bus station, hospital, shopping centres, amusement centres, etc.); the frequency of transportation to the region; the proximity of the hotel to the region's unique and historical places; the city transportation line; and the proximity of the accommodation places to the

hotel swimming areas, social areas, and restaurants. The recommendation criterion, which is the final criterion within the specified criteria, includes the title of positive or negative recommendation, as well as the title of satisfactory price/performance. The criteria and subtitles are given in detail in Table 1.

**Table 1.** Determined criteria and subheadings.

| | ARCHITECTURE |
|---|---|
| **1. CRITERION** | Hotel architecture and hotel condition<br>Room size and usefulness<br>Environmental planning and landscaping<br>Usefulness and novelty status of stores and furniture in hotel and rooms<br>Swimming areas (sea, pool, aqua park, beach, etc.)<br>Social areas (restaurants, bars, spa, sauna, hammam, amphitheatre, mini club, etc.)<br>Easy transportation to all points within the hotel<br>Internet infrastructure and parking facilities |
| | **CLEANLINESS** |
| **2. CRITERION** | General cleanliness of the hotel<br>Cleanliness of rooms (bathroom, toilet, linen, etc.)<br>Cleanliness of environment and landscaping<br>Cleanliness of swimming areas (sea, pool, aqua park, beach, etc.)<br>Cleanliness of social areas (restaurants, bars, spa, sauna, hammam, amphitheatre, mini club, etc.)<br>Cleanliness of all tools and equipment used throughout the hotel |
| | **FOOD AND BEVERAGE** |
| **3. CRITERION** | Food and beverage variety<br>Food and beverage taste<br>Food and beverage freshness<br>Brand quality in food and beverages<br>Presentation quality and service speed in food and beverages<br>Easy access to the food and beverage areas within the hotel<br>Kids restaurant and kids' menu<br>Mini bar |
| | **STAFF/SERVICE** |
| **4. CRITERION** | Personnel attention and professionalism<br>Service speed and quality<br>Problem solving speed<br>Check-in/check-out, room service, and lobby services<br>Effective use of foreign language for foreign tourist portfolio<br>Smiling face, kindness, and respect<br>Welcoming and farewell<br>Bell boy service<br>Doctor and medical services |
| | **ENTERTAINMENT/ANIMATION** |
| **5. CRITERION** | Entertainment (party, concert, DJ performances, live music, etc.)<br>Animation (sea, pool, aquapark, beach, etc.)<br>Mini Clubs (kids' activities and child care)<br>Spa, sauna, hammam, amphitheatre, and gym<br>Activities (parachute, balloon, jet ski, games, and competitions, etc.)<br>Local tours, excursions, and organizations |
| | **LOCATION/TRANSPORTATION** |
| **6. CRITERION** | Proximity of the hotel to locations (airport, bus terminal, hospital, shopping centres, entertainment venues, etc.)<br>Frequency of transportation to the place<br>Proximity to urban transportation lines<br>Proximity of the hotel to the regional and historical places<br>Proximity of in-hotel-accommodation to swimming areas and social spaces |
| | **RECOMMENDATION** |
| **7. CRITERION** | Positive recommendation<br>Negative recommendation<br>Positive price/performance recommendation<br>Negative price/performance recommendation |

*3.3. Methodology*

In the study, the LLC (Levin, Lin, and Chu) unit root test developed by Levin et al. (2002) and the IPS (Im, Pesaran, and Shin) unit root test of Im et al. (2003) are performed in order to test the stability of the data obtained within the scope of the research. Then, in order to determine the model to be used in the panel regression analysis, the preliminary tests to be used are determined, and finally analyses are made to estimate the coefficient.

3.3.1. LLC Unit Root Test

According to the LLC unit root test (Levin et al. 2002), which suggests that individual unit root tests have a limited power to the alternative hypothesis, and that there will be highly persistent deviations from the equilibrium, the null hypothesis of this study includes the unit root for each individual time series, whereas the alternative hypothesis was always the result of being stationary for the time series. According to these results, the basic equation of the test is as follows:

$$\Delta y_{it} = \delta y_{it-1} + \sum_{L=1}^{P_i} \theta_{iL}\Delta y_{it-L} + \propto_{mi} d_{mt} + \varepsilon_{it}, m = 1, 2, 3. \tag{2}$$

$d_{mt}$ in equation represents the deterministic variable and $\propto_{mi}$ the coefficient vector. As the value of $P_i$ in the equation is unknown, a three-step procedure is applied while the test is being calculated. In the first step, the ADF (Augmented Dickey Fuller) regression is applied separately for each series in the panel. In the second step, long-term and short-term standard error rates are estimated for each series, and in the last step, the pooled $t$-statistic is calculated.

In the first step, the basic hypothesis given above for each cross section is applied. Lag length ($p_i$) is allowed to vary between the cross sections. For period $T$, the maximum lag length ($p_{max}$) is selected. Preferably, if a smaller lag length is chosen, the $t$-statistic of $\theta_{iL}$ is used. These $t$-statistics have a standard normal distribution based on a null hypothesis.

After $P_i$ is determined, $\Delta y_{it-L}$ and $d_{mt}$ are applied to the appropriate deterministic variable $\Delta y_{it}$ and $y_{it-1}$ regressions in order to obtain $e_{it}$ ve $v_{it-1}v_{it-1}$ residues. These residuals are normalized using the $e_{it} = \frac{e_{it}}{\sigma_{\varepsilon i}}$ ve $v_{it-1} = \frac{v_{it-1}}{\sigma_{\varepsilon i}}$ calculations. In the second step, long-term and short-term standard error rates are calculated. Based on the null hypothesis that accepts the unit root, the long-run variance of the model is calculated by (Levin et al. 2002), as follows:

$$\sigma_{yi}^2 = \frac{1}{T-1}\sum_{t=2}^{T}\Delta y_{it}^2 + 2\sum_{L=1}^{K} w_{KL}\left[\frac{1}{T-1}\sum_{t=2+L}^{T}\Delta y_{it}\Delta y_{it-L}\right]. \tag{3}$$

In this equation, $K$ is the transition lag and $L$ is the normal lag. $K$ should be obtained so as not to impair the consistency of the variance. The Barlett kernel is calculated using the formula $w_{KL} = 1 - \left(\frac{L}{K+1}\right)$. The formula $S_N = \frac{1}{N}\sum_{i=1}^{N} s_i$ is used to calculate the average standard error.

In the third step, the panel test statistics are calculated using the following regression generated by the number of $NT$ observations:

$$e_{it} = \rho v_{i,t-1} + \varepsilon_{it} \tag{4}$$

In the equation, $t$ represents the average number of observations for each cross section and $\rho$ represents the average lag length of the individual ADF (Çetin and Ecevit 2010).

3.3.2. IPS Unit Root Test

According to this test, which allows the heterogeneity of the cross sections and is calculated as the average of the individual unit root test statistics, the regression equation is as follows (Baltagi 2011; Im et al. 2003):

$$\Delta y_{it} = \mu_i + \beta_i y_{i,t-1} + \sum_{k=1}^{p_i} \theta_{i,k} \Delta y_{i,t-k} + \gamma_i t + \varepsilon_{it} \tag{5}$$

The equation is fixed and trendy. Therefore, it is necessary to subtract the trend from the equation in order to obtain the fixed equation. According to the IPS test, the rejection of the null hypothesis means that one or more of the series are stationary.

Im et al. (2003) calculated the *t*-statistic for each cross section as $t_i = \beta_i / sh(\beta_i)$. Then, by taking the average of $t_i$, the Z statistic is calculated with the following formula:

$$Z = \left( \frac{\sqrt{N}(t - E(t))}{var(t)} \right) \sim N(0,1) \tag{6}$$

The *t* value in this form is obtained by (Çetin and Ecevit 2010), as follows:

$$t = \frac{1}{N} \left( \sum_{i=1}^{N} t_i \right) \tag{7}$$

## 4. Results

The study includes the empirical analysis of the Mediterranean Region, which consists of 25 hotels, in order to determine the factors affecting the hotel preferences of customers and the effectiveness levels of these factors. In this context, firstly the stationary of the series is examined through the panel unit root test. Then, the preliminary tests required to select between the panel fixed and random effects predictors are given, and finally, the coefficients indicating the effect of each independent variable on the hotel preference are calculated.

### 4.1. Unit Root Test Results

While the stability of the series is being tested within the model established for the Mediterranean Region, the unit root test, which is developed by Levin et al. (2002) and Im et al. (2003) is used and the results of unit root test are given in Table 2.

**Table 2.** Unit root test results for the Mediterranean region.

| Variable | LLC | IPS |
|:---:|:---:|:---:|
| *lnSP* | −4.009 * (0.000) | −9.350 * (0.000) |
| *lnARC* | −2.383 * (0.008) | −3.836 * (0.000) |
| *lnGC* | −4.891 * (0.000) | −8.882 * (0.000) |
| *lnFB* | −4.056 * (0.000) | −7.504 * (0.000) |
| *lnPS* | −2.903 * (0.001) | −7.639 * (0.000) |
| *lnEA* | −3.697 * (0.000) | −7.268 * (0.000) |
| *lnLOC* | −5.043 * (0.000) | −12.388 * (0.000) |
| *lnREC* | −5.379 * (0.000) | −9.140 * (0.000) |

**Note:** While the LLC (Levin, Lin, and Chu) test was being calculated, the Bartlett Kernel method was used along with the Newey–West bandwidth selection. The maximum lag length is determined depending on the Schwarz information criteria. *, represents statistical significance at a 1% level. IPS—Im, Pesaran, and Shin.

When unit root tests were applied, the Newey–West bandwidth was chosen for the autocorrelation problem, and optimum lag lengths were determined according to the Schwarz information criteria (SIC). When the results in Table 2 are examined, it is observed that the null hypothesis "series contains unit root" for all of the variables is rejected by both tests, and therefore the variables are stationary in the level values. This finding has led to the use of the panel regression method, using the level values of the coefficient estimates series, as it conflicts with the situation required for the search for the cointegration relationship between the variables. This situation can be interpreted as that the comments made on the TripAdvisor site are not affected by each other, and the data may be static.

### 4.2. Preliminary Test Results

Prior to obtaining coefficient estimates in the panel data analysis, it is necessary to apply preliminary tests in order to choose between pooled regression, fixed effects, or random effects in the model to be used. For this reason, the F-test was used primarily to make a choice between fixed effects and pooled regression. When the F-test results in Table 3 are examined, a conclusion has been reached that the null hypothesis, which emphasizes the necessity of using pooled regression, is rejected, so that the fixed effects should be preferred for the pooled regression.

**Table 3.** F-test results.

|  | Statistic | d.f. | Prob. |
|---|---|---|---|
| Cross-section F | 62.540 | (24,868) | 0.000 |
| Cross-section Chi-square | 903.620 | 24 | 0.000 |

After determining that the fixed effect model should be preferred to pooled regression, LM tests were applied to choose between the random effects model and the pooled regression. Table 4 shows the results of different LM tests, in which the null hypothesis that the pooled model should be preferred to the random effects model. When the results are examined, it is seen that the zero hypothesis is rejected for all the tests, and therefore the random effects model should be preferred to the pooled model.

**Table 4.** LM test results.

| Tests | Cross-Section | Time | Both |
|---|---|---|---|
| Breusch–Pagan | 5039.448 (0.000) | 10.445 (0.001) | 5049.893 (0.000) |
| Honda | 70.989 (0.000) | 3.231 (0.000) | 52.482 (0.000) |
| King-Wu | 70.989 (0.000) | 3.231 (0.000) | 56.737 (0.000) |
| Standardized Honda | 86.311 (0.000) | 3.340 (0.000) | 52.716 (0.000) |
| Standardized King-Wu | 86.311 (0.000) | 3.340 (0.000) | 58.525 (0.000) |
| Gourierioux et al. | – | – | 5049.893 (<0.01) |

Finally, when the results of the Hausman test (Table 5), used to select between the fixed effects model and the random effects model, are examined, it is concluded that the null hypothesis, which indicates the necessity of using random effects, is rejected, and therefore the fixed effect model is the most accurate model for analysis.

<div align="center"><strong>Table 5.</strong> Hausman test results.</div>

|  | Chi-Sq. Statistic | Chi-Sq. d.f. | Prob. |
|---|---|---|---|
| Cross-section random | 65.684 | 7 | 0.000 |

### 4.3. Coefficient Estimation Results

The results of the panel fixed effect coefficient estimation for the interpretation of the determined factors and the significance ratings are shown in Table 6 in detail.

<div align="center"><strong>Table 6.</strong> Panel fixed coefficient estimation results.</div>

| Variable | Coefficient | Std. Error | *t*-Statistic | Prob. |
|---|---|---|---|---|
| *lnARC* | 0.124 | 0.017 | 6.971 | 0.000 |
| *lnGC* | 0.195 | 0.034 | 5.683 | 0.000 |
| *lnFB* | 0.233 | 0.031 | 7.301 | 0.000 |
| *lnPS* | 0.210 | 0.030 | 6.864 | 0.000 |
| *lnEA* | 0.173 | 0.023 | 7.288 | 0.000 |
| *lnLOC* | 0.251 | 0.037 | 6.686 | 0.000 |
| *lnREC* | 0.092 | 0.033 | 2.749 | 0.006 |
| *C* | −0.375 | 0.042 | −8.762 | 0.000 |
| Cross-section fixed (dummy variables) | | | | |
| R-squared | 0.992 | Mean dependent var | | 1.372 |
| Adjusted R-squared | 0.992 | S.D. dependent var | | 0.146 |
| S.E. of regression | 0.012 | Akaike info criterion | | −5.858 |
| Sum squared resid | 0.140 | Schwarz criterion | | −5.688 |
| Log likelihood | 2668.447 | Hannan–Quinn criter. | | −5.793 |
| F-statistic | 3803.403 | Durbin–Watson stat | | 1.545 |
| Prob(F-statistic) | 0.000 | | | |

It is seen that all of the factors observed when the fixed effect model results for the Mediterranean region are examined, according to the data in Table 6, are positive and statistically significant for the effects of the stars on the mean of the star points. According to the findings, a 1% increase in architectural quality of the hotel is preferred by the hotel, and the average star rating is increased by 0.12%; the increase in the hotel's cleanliness increases the hotel's preference by the customer, and increases the average star rating by 0.19%; a 1% increase in the hotel's food and beverage availability increased the hotel's customer preference and average star rating by 0.23%; the 1% increase in hotel staff-service quality increases the hotel's customer preference and average star rating by 0.21%; a 1% increase in the hotel's entertainment and animation facilities increases the hotel's customer preference and average star rating by 0.17%; a 1% increase in hotel location and transportation increases the hotel's preference by the customer and increases the average star rating by 0.25%; and finally, a 1% increase in tastes shows that the hotel is preferred by the customer and the average star rating is increased by 0.09%.

## 5. Discussion and Conclusions

When the findings are evaluated, it is found that the hotels in the Mediterranean region are preferred by the customers, and in particular, the transportation facilities to the locations and the places where the hotels are located are more important than the other factors of the food and beverage concept, the quality of personnel–service, cleaning elements, and entertainment–animation programs. On the other hand, it is possible to say that the recommendation and architectural structure of the hotels by the other customers is relatively less important than the hotel preferences in the Mediterranean region.

The Mediterranean region, because it is both in the position of Turkey's coastline and because of its geographical features, has an important role in the tourists' hotel preference. As some of the hotels

in this area are located at the seaside, and the historical and unique places of this region are found here, it can be thought that the hotel location has become the foreground from the comments received from the TripAdvisor site. This may also be due to the fact that most of the customers do not want the hotel to be at the sea side of the accommodation.

The Mediterranean coast, Turkey's coastline, because of its location and the presence of the geographical features, also plays an important role in the tourist resort of choice. As some of the hotels in this region are located at the seaside and there are historical and unique places of this region, it can be thought that the hotel location comes to the fore when looking at the comments received from tripadvisor.com.tr. This situation may also be caused by the fact that most of the customers want the hotel to be located right by the sea. Accordingly, the fact that the sea areas of the hotels are blue-flagged and that the hotels have their own natural beauty in their own locations, can cause this criterion to come to the forefront. In addition, it can be reached, that given the disturbances in the transportation resources of some regions, the distances to the transportation points of the hotels, and the disturbances arising from the frequency of transportation, and considering the transportation situation for the Mediterranean region and the proximity to the airport, bus terminal, places with historical and natural beauties, and transportation frequency. In addition to these, it can be concluded that the intra-city transportation line can be an important point in terms of frequency and access to every point. The way in which the location criteria of the visitors emerged from the comments is largely in parallel with the previous studies by Stringam et al. (2010), Limberger et al. (2014), and Xie et al. (2014).

It is thought that it may be because of the fact that another crucial factor, the food and beverage criterion, is important in terms of the Mediterranean region, the fact that the vast majority of the regions have conceptions such as "all inclusive" and "ultra all inclusive", and the fact that hotel guests want to try different cultures while finding food and beverages belonging to the their own culture at the same time. In addition, having a variety of food and beverage, appealing to the customers, and having a variety of mini bars in bars and rooms, can also be considered as reasons for preference of customers. In addition to these, the freshness of the products used, product quality, presentation, and the choice of popular brands can make this criterion important for the customers. In this context, the food and beverage criterion (Barriocanal et al. 2010; O'Connor 2010; Stringam et al. 2010), which is the most prevalent of the previous studies, is also at a high level of importance in this study.

The fact that the personnel–service factor is important is interpreted as the desire of customers to reach these services through professional staff in the five-star and multi-service hotels in the region. It can be thought that it is important that the location and transportation possibilities of the hotels are above certain standards, and that the food and beverage services are offered unlimitedly, as well as the presentation quality, speed, and quality of these services, and also the smiling face of the working staff and quick and effective solutions to the problems. The fact that the staff and service criteria are of great importance in the studies conducted by O'Connor (2010), Stringam et al. (2010), and Limberger et al. (2014) almost coincides with the results of this study.

The fact that the cleaning factor has an important place among other criteria can be shown to be equivalent to the results of other studies in the literature (Stringam et al. 2010; Limberger et al. 2014; Xie et al. 2014). The cleanliness of public areas, such as the general cleanliness of the hotels and rooms; cleanliness of general and special material; environmental cleanliness; swimming areas such as sea, pool, aqua park, beach; and restaurants, bars, spa, sauna, hammam, amphitheatre, and mini club, and that the cleanliness criterion is the foreground.

The entertainment–animation criterion is thought to be among the criteria preferred by customers in terms of importance for both adults and children separately. It is thought that for adults, pool and sea games, various activities, and organized sports programs, as well as special activities such as parties and organizations planned for the night, spa, sauna, and hammam, as well as mini clubs' activities for children during the day and night, various fun activities, and nursing services for children make this criterion important. In addition, it may be concluded that the tours, excursions, and organizations suitable for the region are important for this criterion. The fact that this criterion is not included

in the previous studies is probably thought to be due to the fact that the studies are carried out in city-dwellers and pension-type accommodation places, and that the selected hotel facilities are limited. In this respect, it is expected that the importance of the entertainment–animation criteria for the Mediterranean region will contribute to the literature.

Moreover, location–transportation, food–beverage, personnel–service, and other criteria that are less important than the entertainment–animation criteria come to the fore. This situation is thought to have remained on the second planet, because of the fact that the districts generally have new and spectacular structures and are also located in large areas, as well as the hotel architecture, and the hotel general conditions, environment, and landscape arrangements that are similar to each other.

The beach facilities of the hotels in the region, the swimming areas (sea, pool, aquapark, beach, etc.), social areas (restaurants, bars, spa, sauna, baths, amphitheatre, gyms, mini club, etc.), the size and usefulness of the rooms, the usefulness and novelty of the goods and fixtures in hotels and rooms, the proximity of the swimming and social areas of the hotels, and the internet infrastructure and parking facilities, show that this criterion is considered less important compared with other criteria. In this context, contrary to the studies done in the literature (Barriocanal et al. 2010; Stringam et al. 2010; Limberger et al. 2014; Kirillova et al. 2014; Siamionava et al. 2018), the result of hotel architecture is less important for this region.

Considering the criteria derived from the comments received from the TripAdvisor site, it can be deduced that the recommendation effect is minimized, and that the criteria set by the users, or, in other words, by the potential customers, are influenced by the interpretations reviewed, and it can be deduced that they make their own decision according to these criteria, it may show that they are not affected by inferences like "I recommend" or "I do not recommend" in the comments made. Moreover, it is seen that the price–performance measures under the recommendation effect are not taken into consideration in the literature, and this does not parallel the results in the literature (Barriocanal et al. 2010; O'Connor 2010; Limberger et al. 2014). When this is generally interpreted, it can be explained that the potential customers have taken into account the specified criteria, but made their own decision without depending on the recommendations given.

Based on the results of the study, the following suggestions can be made to hotel managers in the sector: hotel managers may first make recommendations from reviews and evaluations made by previous customers on the site at the TripAdvisor site on which aspects of the business should be prioritized for missing or improved aspects, and they can take steps to improve the criterion or criteria that are seen as incomplete.

Later on, free-of-charge services can be arranged for the customers at the airport and bus terminal, or the frequency of the existing transportation services can be increased for the criteria of location–transportation possibilities, which have much more preference in the consumer preferences in the frame of the determined criteria.

In addition, special tours can be arranged to places specific to the Mediterranean region, in terms of its historical and geographical location. In addition, special boating tours can also be organized to the bays of the region. In terms of the food–beverage criterion, which also holds an important place in the consumer's preferences, the hotels can expand their current concept. In addition, different tastes can be offered to customers by having kitchens catering to different cultures. In addition to this, by paying attention to the quality and freshness of the food and beverage, customer satisfaction can be achieved by attaching importance to brand quality. For the personnel–service and cleaning criteria, besides the fast and high quality service approach, services should be provided with a smiling face, have fast problem-solving, and be completely professional.

For the cleaning criterion, it is possible to increase the customer satisfaction by paying attention to the cleanliness of the hotel and the cleanliness of the environment, as well as the cleanliness of the room, which is the most important detail of the customers, and the cleanliness of all of the used tools and equipment. For the entertainment–animation criterion, which is one of the other criteria in terms

of importance, children's programs as well as programs for adults should be considered and given sufficient importance.

In the study, the criteria determined by the content analysis based on the customer comments on tripadvisor.com.tr may vary for both national and international hotels. The mentioned differences form the limitations of the study in terms of not validating the criteria for each region and every situation. For example, in the Mediterranean region, hotels will have food and beverage concepts such as "all-inclusive" and "ultra-all-inclusive", which will differ in the evaluation of this criterion compared to the "bed and breakfast" hotels in another region. In addition, it will be inevitable that the determined criteria such as the hotel architecture in the Mediterranean region, transportation facilities, location, services provided, cleanliness concepts, and entertainment and animation facilities will differ in the evaluations to be made for other hotels in other regions.

**Author Contributions:** The share of contributions of authors are equal.

**Funding:** This research received no external funding.

**Conflicts of Interest:** The authors declare no conflict of interest.

## Appendix A

**Table A1.** The Mediterranean Region's Hotels and Comments.

| RANK | HOTEL NAME | REGION | NUMBER OF COMMENTS |
|------|------------|--------|--------------------|
| 1 | Voyage Belek Golf and Spa | Belek | 6527 |
| 2 | Julian Club Hotel | Marmaris | 4579 |
| 3 | Rixos Premium Belek | Belek | 4499 |
| 4 | Limak Atlantis Deluxe Hotel and Resort | Belek | 4022 |
| 5 | Rixos Sungate | Kemer | 4011 |
| 6 | TUI BLUE Marmaris | Marmaris | 3946 |
| 7 | Maxx Royal Belek Golf Resort | Belek | 3637 |
| 8 | Club Tuana Fethiye | Fethiye | 3410 |
| 9 | Green Nature Resort and Spa | Marmaris | 3403 |
| 10 | Kaya Belek | Belek | 3287 |
| 11 | Titanic Deluxe | Belek | 3242 |
| 12 | Limak Arcadia Golf & Sport Resort | Belek | 3128 |
| 13 | Club Med Palmiye | Kemer | 2906 |
| 14 | PALOMA Foresta Resort & Spa | Kemer | 2855 |
| 15 | Blue Bay Platinum Hotel | Marmaris | 2849 |
| 16 | Ideal Prime Beach | Marmaris | 2827 |
| 17 | Crystal Tat Beach Golf Resort & Spa | Belek | 2727 |
| 18 | Barut Kemer | Kemer | 2665 |
| 19 | SENTIDO Zeynep Resort | Belek | 2617 |
| 20 | Liberty Hotels Lykia | Fethiye | 2617 |
| 21 | Cornelia Diamond Golf Resort & Spa | Belek | 2513 |
| 22 | Limak Limra Otel | Kemer | 2512 |
| 23 | Club Asteria Belek | Belek | 2492 |
| 24 | SunConnect Grand Ideal Premium | Marmaris | 2395 |
| 25 | Letoonia Club Hotel | Fethiye | 2393 |

**Note:** The number of comments of the hotels determined in the Mediterranean region varies between 6527 and 2393. The first hotel belonging to the Mediterranean region in this frame is the Voyage Belek Golf and Spa Hotel in the Belek region, with 6527 comments. The last place is Letoonia Club Hotel in Fethiye region, with 2393 comments.

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
