# Peer review of "The Determinants of Tourist Preferences of the Mediterranean Region in Turkey"

_admsci, doi:10.3390/admsci8040081_

Round 1

Reviewer 1 Report

The author(s) discuss the determinants of tourist preferences analysing the comments on TripAdvisor about the hotels in the Mediterranean Region. The introduction, the theoretical background and the research findings are well described; the methodology, the discussion and the conclusions need to be revised.

I encourage the author(s) to revise the manuscript as follow:

Section 3: describe the main features of tourism sector in the Mediterranean Region and add data about accommodation (with special reference to the hotel sector) and tourism flows. In the introduction (see rows 66-69) you underlined some pull factors of Turkey (e.g. natural beauties) which are also recalled in the conclusions. Before you introduce the “empirical model” I suggest to add a section to describe the main features of the study area.

Section 3.2 Data Set: you declare that the sample is made of 25 hotels. How many hotel are situated in the study area?

Row 173 refers to “Annex 1”, but at the end of the paper you rename it as “Appendix 1”. Keep the terminology consistent.

To help the reader to read smoothly through your analysis, organise information and description of the 7 criteria (rows 177-203) in a Table.

Section 3.3: revise the text and titles of this sub-section. For example: row 206: you refer to LLC and IPS but the references are found in the following section.

Row 223: you refer to ADF regression but you explain it only in row 247 (Augmented Dickey Fulle - ADF). Revise this section in the light of these observations.

Section 4.3 Coefficient Estimation Results: Row 331: you refer to Table 7, but this Table is not shown in the paper. Correct the table number.

Discussion and conclusions: organise the text in two sessions. Strengthen the theoretical contribution of the study. Reflect about the relevance of the star rating considering also that there is not a unique classification at international/national level (i.e. within a same country different regions might adopt different criteria to classify hotels).

Translate the first row of the Table insert in Appendix 1 (Sira, Otel ADI, etc.).

Finally, look carefully for typo.

Author Response

Thank you very much for your valuable feedback. I have made the necessary corrections. Thank you again. Good work.

Reviewer 2 Report

This is a very interesting and well-written paper, which will attract attention of the international research audience. The only very minor revision is required. My recommendations are as follows.

1) Please, delete TripAdvisor from the title - this is necessaery to make the title less technical

2) Please, explain in Introduction, what does the abbreviation WOM mean.

3) Line 83: 2000 -> 2000s

4) Line 84: US -> USA

5) 4. EMPIRICAL FINDINGS -> RESULTS

6) Line 365: delete comma after frequency

7) I encourage to discuss your results briefly in regard to the importance of design and aesthetics of hotels to their perception. Please, refer to these works:

https://www.sciencedirect.com/science/article/pii/S0261517713002185

https://www.sciencedirect.com/science/article/pii/S0278431917302062

8) Finally, check, please, whether citations and section headings are given strictly according to the journal rules. And why journal volumes are not indicated in References?

Author Response

(The authors gave the same response as above.)

Round 2

Reviewer 1 Report

The author revised the manuscript following the suggestions and he/she answered the comments of the referee. I don’t have other observations about the paper.